# Nanoparticle-Based Radioconjugates for Targeted Imaging and Therapy of Prostate Cancer

**DOI:** 10.3390/molecules28104122

**Published:** 2023-05-16

**Authors:** Anna Lankoff, Malwina Czerwińska, Marcin Kruszewski

**Affiliations:** 1Centre for Radiobiology and Biological Dosimetry, Institute of Nuclear Chemistry and Technology, Dorodna 16, 03-195 Warsaw, Poland; m.kruszewski@ichtj.waw.pl; 2Department of Medical Biology, Institute of Biology, Jan Kochanowski University, Uniwersytecka 15, 25-406 Kielce, Poland; 3Department of Dietetics, Institute of Human Nutrition Sciences, Warsaw University of Life Sciences (WULS-SGGW), 159c Nowoursynowska, 02-776 Warsaw, Poland; m.czerwinska@ichtj.waw.pl; 4Department of Molecular Biology and Translational Research, Institute of Rural Health, Jaczewskiego 2, 20-090 Lublin, Poland

**Keywords:** prostate cancer, prostate cell surface receptors, nanoparticles, radionuclides, nanoparticle-based radioconjugates, targeted imaging, targeted therapy, theranostics

## Abstract

Prostate cancer is the second most frequent malignancy in men worldwide and the fifth leading cause of death by cancer. Although most patients initially benefit from therapy, many of them will progress to metastatic castration-resistant prostate cancer, which still remains incurable. The significant mortality and morbidity rate associated with the progression of the disease results mainly from a lack of specific and sensitive prostate cancer screening systems, identification of the disease at mature stages, and failure of anticancer therapy. To overcome the limitations of conventional imaging and therapeutic strategies for prostate cancer, various types of nanoparticles have been designed and synthesized to selectively target prostate cancer cells without causing toxic side effects to healthy organs. The purpose of this review is to briefly discuss the selection criteria of suitable nanoparticles, ligands, radionuclides, and radiolabelling strategies for the development of nanoparticle-based radioconjugates for targeted imaging and therapy of prostate cancer and to evaluate progress in the field, focusing attention on their design, specificity, and potential for detection and/or therapy.

## 1. Introduction

Prostate cancer (PCa) is the second most frequent malignancy in men worldwide and the fifth leading cause of death by cancer, with a near estimate of 1.4 million new cases and 375,000 deaths worldwide [1,2]. According to the latest statistical data from the Global Cancer Observatory 2020 (GLOBOCAN), owned by the World Health Organization/International Agency for Research on Cancer, incidence rates of prostate cancer vary from 6.3 to 83.4 per 100,000 men across regions with the lowest rate in South Central Asia and the highest rate found in Northern Europe (Ireland). Regional patterns of mortality rates do not follow those of incidence, with the highest mortality rate in the Caribbean (27.9). Established risk factors for prostate cancer are limited to advancing age, family history of this cancer, lifestyle and environmental factors, and certain genetic mutations [1]. Five-year survival in patients with localised PCa is nearly 100%, but after primary treatment with radical prostatectomy, approximately one-third of the patients will present biochemical recurrence [3]. Although most patients with biochemical recurrence initially benefit from systemic therapy, many of them will progress to non-metastatic castration-resistant prostate cancer (nmCRPC) or metastatic castration-resistant prostate cancer (mCRPC), i.e., still incurable. The reported median survival across different studies ranges from 9 to 30 months in mCRPC patients [4]. The significant mortality and morbidity rate associated with the progression of this disease results mainly from a lack of specific and sensitive prostate cancer screening systems, identification of the disease at mature stages, and failure of anticancer therapy [5].

Conventional imaging modalities available for the detection of localised PCa, locally recurrent PCa or distant metastatic spreads have traditionally included ultrasound imaging (USI), computed tomography (CT), magnetic resonance imaging (MRI), and ^99m^Tc-labelled bisphosphonate bone scintigraphy (BS) [6]. Although widely available, these modalities have significant disadvantages, including limited tissue contrast between cancerous and benign tissue, poor resolution, moderate specificity, and limited accuracy to detect prostate cancer metastases [7]. Development in imaging technologies and combination of different imaging systems, e.g., positron emission tomography with CT (PET/CT), single-photon emission computed tomography with a low-dose CT (SPECT/CT), multiparametric MRI (mpMRI), and its fusion with trans-rectal ultrasound-guided biopsy (mpMRI-TRUS) or with ultrasound imaging (mpMRI–US), have vastly improved the preciseness of detection and restaging of recurrent PCa. Despite significant benefits derived mainly from the simultaneous registration of anatomical and functional data within a single study, these techniques have also some disadvantages. Among them, the most critical limitation is the poor detection of macrometastases (>2 mm) and micrometastases (0.2–2 mm) [7,8,9,10].

Over the last years, much attention has been paid to prostate-specific membrane antigen (PSMA) as a target for personalised imaging and therapy of PCa. The PSMA is a cell surface receptor with a large extracellular domain that allows for effective targeting. The PSMA is overexpressed on prostate cancer cells and in the neovasculature of several types of solid tumours. Its expression increases with the progress of the disease and its metastatic status [11]. The PSMA is also expressed in normal healthy tissues, but its expression is significantly smaller [12].

Several PSMA ligands have been developed so far, such as small-molecule inhibitors, antibodies, antibody-based molecules and aptamers, and some of them have been transferred to the clinic [13,14]. To date, nearly 450 clinical trials involving PSMA-targeted PCa imaging and/or therapy have been completed, are ongoing, or have been approved worldwide. According to the ClinicalTrials database (https://clinicaltrials.gov (accessed on 1 March 2023), the vast majority of these studies have been dedicated to PSMA-targeted radiopharmaceuticals for PET, SPECT, and PSMA-guided radiotherapy. Currently, the PSMA PET imaging utilizing [^68^Ga]Ga-PSMA-11 or [^18^F]F-DCFPyL is approved by the US Food and Drug Administration (FDA) and recommended in the Guidelines of the European Association of Urology as an imaging technique of choice for the majority of patients with biochemically recurrent PCa after curative treatment and for assessment of metastases [15,16]. However, its value for the primary diagnosis and staging of PCa is still under debate [17]. Satisfactory results obtained with the PSMA PET radiopharmaceuticals and wider distribution of SPECT technology over PET have prompted the development of PSMA SPECT tracers, focused on technetium-99m (^99m^Tc). Over the past years, a number of different [^99m^Tc]Tc-PSMA agents have been explored for PCa imaging and radio-guided surgery. Although clinical studies have shown that these molecular probes possess high sensitivity and specificity in the initial diagnosis of PCa, monitoring of disease progression and response to therapeutic treatment, none of them exceeds the accuracy and sensitivity of the PSMA PET radiopharmaceuticals [18,19]. It is believed that multimodal nanoparticles (NPs) for PCa-targeted imaging can offer synergistic advantages over any modality alone and can be a promising cancer imaging in the future.

Apart from targeting the PSMA receptor for diagnostic purposes, there is increasing interest in the use of PSMA ligands combined with radionuclides for therapeutic applications in mCRPC patients, who have failed standard treatments with curative intent (e.g., the first- and second-generation antiandrogens, taxane-based chemotherapies, immunotherapy, and radium-223-chloride) [15,20]. The PSMA-targeted radioconjugates that have been most commonly evaluated for therapy of end-stage mCRPC patients are mainly based on β^−^-emitting lutetium-177 and α-emitting actinium-225 [13,21]. In general, clinical experience from these studies has shown excellent tumour targeting, satisfactory pharmacokinetic properties, prolonged overall survival, and moderate adverse events. In March 2022, [^177^Lu]Lu-PSMA-617 (PluvictoTM) was approved by the FDA for the treatment of patients with PSMA-positive mCRPC who have previously failed other therapy options [22]. Despite the benefits of PSMA-targeted radionuclide therapy, there are also some limitations. According to a recent meta-analysis, around 40% of patients are resistant to the treatment with [^177^Lu]Lu-PSMA-617 [23]. Moreover, therapy with [^177^Lu]Lu-PSMA-617 is not efficient for the treatment of micrometastatic disease and circulating tumour cells because β^−^ particles emitted from ^177^Lu have a relatively long range in the soft tissue (1–10 mm), well beyond a diameter of micrometastases [24]. Radiotherapy with α-emitting radionuclides seems to be a better alternative to this based on β^−^ particles because α particles deposit their whole energy within a few cell diameters (<100 µm), efficiently inducing DNA damage [25]. However, one of the most important obstacles associated with α-emitting radionuclides is the liberation of the recoiled daughter radionuclides rising during decay, which allows them to freely migrate in the body, causing significant toxicity to healthy tissues and decreasing the therapeutic dose delivered to the tumour [26,27]. Encapsulation of radionuclides in tumour-targeting NPs may assist in overcoming problems associated with the harmful consequences of α-radionuclide therapy.

The purpose of this review is to discuss briefly the selection criteria of suitable NPs, ligands, radionuclides and radiolabelling strategies for the development of NP-based radioconjugates for targeted imaging and/or therapy of PCa and to evaluate major developments in this field, focusing attention on their design, specificity, and potential for detection and/or therapy.

## 2. Selection Criteria of Suitable NPs, Ligands, Radionuclides, and Radiolabelling Strategies for Development of NP-Based Radioconjugates for Targeted Imaging and Therapy of PCa

In parallel with the development of PSMA-targeted radiopharmaceuticals, nanotechnology-based delivery systems with diverse payloads have been designed for their potential use in Pca-targeted imaging, therapy, and theranostic applications [28,29]. Among them, NPs labelled with radionuclides have gained increasing attention for their use in nuclear medicine applications. However, the success of targeted cancer imaging and therapy using radiolabelled NPs relies upon the choice of a suitable (1) type of NPs, (2) type of ligand (targeting vector), (3) type of radionuclide, (4) radiolabelling strategy, and (5) stability of radiolabelled conjugate.

### 2.1. Selection of a Suitable Type of NPs for Targeted Imaging and Therapy of PCa

So far, different types of NPs have been synthesized for their potential use in the PCa-targeted nuclear nanomedicine, including silica NPs [30], gold NPs [31,32,33], liposomal NPs [34,35,36], texaphyrin NPs [37], polymer NPs [38,39,40], micellar NPs [41,42,43], melanin NPs [44], copper sulphide NPs [45], gadolinium vanadate NPs [46], iron oxide NPs [47,48], quantum dots (QDs) [33,49], and zeolite NPs [50] (Table 1). These NPs were characterised via various techniques, such as transmission electron microscopy (TEM), scanning electron microscopy (SEM), dynamic light-scattering method (DLS), X-ray powder diffraction (XRD), UV-visible spectroscopy, thermogravimetric analysis (TGA), inductively coupled plasma–atomic emission spectroscopy (ICP-AES), FT-IR analysis, etc. Physicochemical analyses revealed that although these NPs differ in composition, they share the most important attributes affecting their interaction with biological systems, such as small spatial dimensions and shape. Microscopic analyses showed that the size of all nanoparticles ranges between 5 and 180 nm. In addition, almost all NPs exhibit spherical morphology (with the exception of two types of nanoparticles: tetragonal sheet-like gadolinium vanadate NPs and 4-armed starPEG40kDa NPs). Other techniques of NP characterization revealed their good dispersibility, stability, and high feasibility of their surface for modification and functionalization. A detailed physicochemical characterization of these NPs has been described in the literature referred to above. In the design of efficient NPs for targeted imaging, therapy, and theranostic applications, these properties of NPs have been identified as the most important factors. The size of NPs determines their ability to translocate across tissues and organs, cross biological barriers, and enter cells and affects their pharmacokinetics, biodistribution, and tumour penetration. The “ideal” size for the receptor-targeted NPs as carriers for diagnostic and therapeutic agents ranges from 10 to 60 nm, regardless of the NP’s composition and surface charge [51]. NPs with sizes greater than 100–200 nm are easily identified by macrophages, leading to their accumulation in the elements of the reticuloendothelial system, such as the liver, spleen, lungs, and bone marrow. In contrast, NPs with sizes less than 10 nm are rapidly cleared by the renal excretion system. Moreover, NPs smaller than 10 nm have too small of a surface area to interact with cell membrane receptors, which can result in decreased cellular uptake. On the other hand, NPs with sizes greater than 60 nm can cause steric hindrance and receptor saturation [52]. In addition to their size, the shape of NPs also plays an important role in cellular uptake, persistence in blood circulation, and biodistribution. In vitro and in vivo studies revealed that spherical NPs exhibit the fastest internalization rate, followed by cubic NPs, rod-like NPs, and disk-like NPs [53].

The chemical properties of the surface of NPs and their affinity towards different chemical groups define the ability of NPs to bind different molecules in the process of radioconjugate preparation. Since modification of the surface of NPs for biomedical purposes was reviewed elsewhere in detail [54], here, we would give only a general overview of this topic. In the first step of the surface modification, homo- or hetero-bifunctional crosslinkers are used in order to add various organic functional groups. For example, the surface of silica NPs or zeolite NPs can be modified via aminosilanes that provide amino groups useful for subsequent conjugation with compound-of-interest [55]. The surface of gold NPs is usually modified with crosslinkers with -SH or -NH_2_ groups that are able to produce a covalent bond with the metal [56]. The surface of liposomes or micelles can be modified via the introduction of polyethylene glycol (PEG) to lipid anchor [57]. For metal oxides and quantum dots (QDs), the most-used strategy is based on the substitution of the original chemical moieties present on the NP’s surface with functional groups of interest, such as a diol, thiol, amine, or carboxylic acid [58]. These “first-step” functional groups can be further linked with mono- or bifunctional PEG molecules, spacers, mono- or bifunctional chelators, radionuclides, or receptor-targeting ligands.

**Table 1 molecules-28-04122-t001:** Nanoparticle-based radioconjugates for targeted imaging and therapy of prostate cancer.

NPs	NPs Size	Radionuclide	Ligand	Final Compound	Modality	Cell Type/Animal Models	References
gadolinium vanadate NPs (GdVO4)	∼150 nm	Copper-64	Asp-Gly-Ala (DGEA) peptide	^64^Cu-DOTA-GdVO 4 :4%Eu-DGEA	PET/MR	prostate cancer PC-3 cells, athymic nude mice bearing PC-3 xenograft	[46]
micellar NPs (LNP)	12 nm	Copper-64	single chain (scFv)	^64^Cu-DOTA-scFv-LNP	PET	NOD/SCID mice bearing LNCaP xenograft	[42]
copper sulfide NPs (CuS)	5 nm	Copper-64	Bombesin (7–14)	Bom-PEG-[^64^Cu]CuS	PET	prostate cancer PC-3-KD1 cells Nu/Nu mice bearing PC-3-KD1 xenograft	[45]
iron oxide NPs (IO)	11 nm	Gallium-68	glutamate-urea-lysine ligand	^68^Ga-DOTA-IO-GUL	PET/MR	prostate cancer 2Rv1, LNCaP, and PC-3 cells BALB/c nude mice bearing 2Rv1 and PC-3 xenograft	[47]
iron oxide NPs (mNP-S1/2) (mNP-N1/2)	55–138 nm	Gallium-68	glutamate-urea-lysine ligand Bombesin (7–14)	^68^Ga-mNP-N1/2 ^68^Ga-mNP-S1/2	PET/MRI	prostate cancer LNCaP and PC-3 cells	[48]
quantum dots (QDs)	~12 nm	Fluorine-18	RGD peptide Bombesin (7–14)	^18^F-FP-QD-RGD-BBN	PET/NIFR	prostate cancer PC-3 cells BALB/c nu/nu nude mice bearing PC-3 xenograft	[49]
melanin NPs (MNPs)	13 nm	Iodine-124	glutamate-urea-lysine ligand	^124^I-MNPs-PEG-TL	PET	prostate cancer LNCaP and PC-3 cells athymic nude mice bearing LNCaP and 22RV1 xenograft	[44]
polymer PEG NPs (PEG-(DFB)1) (PEG-(DFB)3)	15 nm	Zirconium-89	ACUPA	^89^Zr-PEG-(DFB)3(ACUPA)1 ^89^Zr-PEG-(DFB)1(ACUPA)3	PET	prostate cancer PC3 and PC3-Flu cells nu/nu athymic mice bearing PC3 and PC3-Flu xenograft	[40]
micellar NPs (CCPM)	~22 nm	Indium-111	TNYL-RAW	^111^In-TNYL-RAW-CCPM(Cy-7)	SPECT/NIRF	prostate cancer PC3-MM2 cells nude mice bearing PC3-MM2 xenograft	[41]
poly(lactic acid)− polyethyene glycol NPs (PLA−PEG)	~100 nm	Indium-111	ACUPA	^111^In-DOTA-RDye680RD-PEG-PLA-ACUPA RDye680RD-PEG-PLA-ACUPA	SPECT/NIFR	prostate cancer PC-3 and PC-3 flu cells athymic mice bearing PC-3 and PC-3 flu dual xenograft	[39]
gold NPs (AuNP)	~20 nm	Technetium-99m	Bombesin (7–14)	^99^mTc-EDDA/HYNIC-GGC-AuNP-Lys3-bombesin	SPECT	prostate cancer PC-3 cells nude mice bearing PC-3 xenograft	[31]
quantum dots (QDs) gold NPs (AuNP)	~6 nm 7 and 14 nm	Technetium-99m	glutamate-urea-lysine ligand	^99m^Th-DAP-HS-PEG (12)-AuNPs-HS-PEG-DAP-TF	SPECT	prostate cancer LNCaP cells nude NMR mice bearing LNCaP xenograft	[33]
gold NPs (DTDTPA-AuNP)	113 nm	Gallium-67	Bombesin (7–14)	^67^Ga-DTDTPA-AuNP-BBN	SPECT	prostate cancer PC-3 cells athymic nude mice bearing PC3 xenograft	[32]
liposomal NPs (LNP)	107 nm	Actinium-225	antibody J591aptamer (A10)	^225^Ac-LNP-PEG-J591 ^225^Ac-LNP-PEG-A10	therapy	prostate cancer LNCaP and Mat-Lu cells, endothelial HUVEC, BT474, and breast cancer MDA-MB-231 cells	[34]
liposomal NPs (LNP)	107 nm	Actinium-225	antibody J591 glutamate-urea-lysine ligand	^225^Ac-LNP-PEG-anti PSMA mAb ^225^Ac-LNP-PEG-GUL	therapy	endothelial HUVEC (PSMA+), HUVEC (PSMA-) and breast cancer MDA-MB-231 cells	[35]
zeolite NPs	~120 nm	Radium-223	antibody D2B	^223^RaA-silane-PEG-D2B	therapy	prostate cancer LNCaP C4-2, DU145 and prostate normal RWPE-1, HPrEC cells BALB/c nude mice bearing LNCaP C4-2 xenograft	[50,59]
curcumin-containing poly(lactic-*co*-glycolic acid) NPs (PLGA-CUR)	76 nm	Iodine-131	antibody J591	^131^I-PSMA-PLGA-CUR	theranostic SPECT/ chemotherapy/ radiotherapy	prostate LNCaP C4-2, DU145 and PC-3 cells athymic nude mice bearing LNCaP C4-2 xenograft	[38]
sorafenib-containing silica NPs (PSi)	~10 nm	Indium-111	iRGD peptide	^111^In-PSi-Alexa488-DBCO-DOTA-iRGD	theranostic SPECT/ IF/ chemotherapy	prostate cancer PC3-MM2 cells Hsd:NMRI-Foxnlnu/nu nude mice bearing PC3-MM2 xenograft	[30]
micellar NPs (LNP)	20 nm	Copper-64	diabody (cys-DB) based on the J591 antibody	^64^Cu-DOTA-cysDB-LNP Dox-DOTA-LNP	theranostic PET/ chemotherapy	NOD/SCID mice bearing LNCaP xenograft	[43]
texaphyrin NPs (texaphyrin)	~100 nm	Indium-111 Lutetium-175	glutamate-urea-lysine ligand	^111^In/^175^Lu-texaphyrin-TL	theranostic SPECT/NIRF/ PDT therapy	prostate cancer PC3 ^luc6^ cells athymic nude mice bearing PC-3 or PC3 flu or PC3 ^luc6^ xenograft	[37]
doxorubicin-containing liposomal NPs modified with P3-liposomes)	~180 nm	Technetium-99m	glutamate-urea-lysine ligand	^99m^Tc-P3-Liposomes	theranostic SPECT/ chemotherapy	prostate cancer LNCaP and PC-3 cells	[36]

### 2.2. Selection of a Suitable Type of Ligand for Targeted Imaging and Therapy of PCa

Through the past decade, a variety of receptor-targeting ligands has been systematically evaluated for their ability to bind to the membrane receptors exclusively expressed in PC tissue and/or highly expressed in PC-dependent metastases, while minimally expressed on non-malignant tissue, and that are accessible to imaging and therapeutic modalities. Among them, the most widely used ligands for NP-based imaging and therapy of PCa are PSMA-targeting small molecules [33,35,36,37,39,40,44,47,48], antibodies [34,35,38,50], antibody single-chain fragments (scFv) [42], diabodies (cys-DB) [43], and aptamers [34]. A comprehensive overview of different classes of PSMA ligands in preclinical and clinical use has been described in our previous paper [13].

In addition to ligands targeting the PSMA receptor, alternative ligands targeting other receptors on PCa cells have been of major interest for NP-based targeted imaging and therapy of PCa, such as bombesin-like peptides, which target gastrin-releasing peptide receptor (GRPR) [31,32,45,49]. The GRPR is highly expressed in prostatic intraepithelial neoplasias and primary and invasive PCa, whereas its expression in normal prostate tissue and benign prostate hyperplasia is relatively low [60]. Bombesin (BBN), a 14 amino acid peptide, and its analogues based on this 14-amino acid backbone sequence (e.g., BBN 6–14) have been investigated as targeting ligands for diagnosis and therapy of GRPR-positive tumours using several different radionuclides [61,62].

Another group of ligands consists of proteins containing tripeptide L-arginine-glycine-L-aspartic acid (RGD) [30,46]. These ligands can specifically recognise an integrin αvβ3 receptor that is highly expressed in many solid tumours, but rare in normal tissues [63]. In PCa, the integrin αvβ3 receptor mediates adhesion, invasion, immune escape, and neovascularization through interactions with different ligands [10]. Several linear and cyclic RGD peptides (e.g., iRGD) have been evaluated so far as radiotracers for imaging and therapy of PCa patients [9,64].

Yet, another group consists of ligands targeting Ephrin receptor B4 (EphB4) [41]. Overexpression of the EphB4 receptor is observed in most stages of PCa and is retained after exposure to androgen deprivation therapy, however is not commonly expressed in normal and benign prostate tissue [65]. The crystal structure of the EphB4 receptor in complex with phage display derived ligands revealed that TNYLFSPNGPIARAW (TNYL-RAW) peptide, which bound to the ephrin-binding cavity of the receptor, was its most promising ligand [66].

### 2.3. Selection of a Suitable Type of Radionuclide for Targeted Imaging and Therapy of PCa

The selection criteria of a suitable radionuclide for the labelling of NPs for imaging and therapy of PCa depend on several factors. In general, the most important factors comprise nuclear decay characteristics, type of emitted energy, half-life, the chemistry of radionuclide, and its usefulness in the chosen radiolabelling strategy.

The nuclear decay properties of radionuclides and the type of emitted energy determine whether a radiopharmaceutical can be used as an imaging agent or as a therapeutic agent. Diagnostic radiopharmaceuticals require radionuclides that emit positrons (β+) or gamma (γ) rays. Positron-emitting radionuclides (e. g. ^64^Cu, ^68^Ga, ^67^Ga, ^18^F, ^89^Zr, ^124^I, etc.) are useful for PET. Gamma-ray-emitting radionuclides (e. g. ^111^In, ^99m^Tc, etc.) are useful for SPECT [67]. In contrast, therapeutic radiopharmaceuticals require radionuclides that emit α particles (e.g., ^225^Ac, ^223^Ra, ^131^I, etc.), β− particles (e.g., ^177^Lu, ^90^Y, etc.) or Auger electrons (e.g., ^161^Tb, ^125^I, etc.) [13]. The properties of radionuclides used in the NP-based agents for targeted imaging and therapy of PCa are shown in Table 2.

In addition to the type of emitted radiation, a half-life of radionuclide decay should be considered. In general, for diagnostic purposes, the half-life of a radionuclide should be short enough to limit the radiation dose and to decay quickly after diagnosis. For therapeutic purposes, the half-life of a radionuclide should be long, as a short decay period would decrease the therapeutic efficiency of a radiopharmaceutical. Moreover, radionuclides selected for imaging and therapy should match the pharmacokinetics of NPs with targeting vectors, which determine the time of deposition of the maximum amount of energy within a target tissue. For example, peptides, small molecules, and antibody-based molecules with fast clearance shuld be labelled by radionuclides with a short half-life. Intact antibodies with slow turnover might be labelled by radionuclides with a long half-life [68]. In the case of therapeutic radiopharmaceuticals, the range of the radiation emitted, an appropriate linear energy transfer value, high radionuclide purity, high radiochemical purity, and high specific radioactivity are also important selection criteria [69,70].

**Table 2 molecules-28-04122-t002:** Characteristics of radionuclides used in NP-based targeted imaging and therapy of PCa. β^−^—beta electrons; β^+^—beta positrons; α—alpha particles; EC—Energy Capture; IT—Isomeric Transition.

Radionuclide	Half-Life	Decay Energy [MeV]	Decay Mode	Production Mode	Reference
**Positron-emitting radionuclides**					
Copper-64	12.7 h	β^+^ 0.653; β^−^ 0.579	β^+^/β^−/^EC	^64^Ni(p,n)^64^Cu	[71]
Gallium-68	67.6 min	β^+^ 1.899	β^+^/EC	^68^Ge/^68^Ga	[72]
Fluorine-18	110 min	β^+^ 0.634	β^+^/EC	^18^O(p,n)^18^F	[73]
Iodine-124	4.17 d	β^+^ 0.819; γ 0.603	β^+^/EC	^124^Te(p,n)^124^I	[74]
Zirconium-89	78.4 h	β^+^ 0.511; 0.902; 0.909	β^+/^EC	^89^Y(p,n)^89^Zr	[75]
**Gamma-emitting radionuclides**					
Indium-111	2.8 h	γ 0.171; 0.245	EC	^111^Cd(p,n)^111m.g^In ^112^Cd(p,2n)^111m.g^In	[76]
Technetium-99m	6.0 h	γ 0.141	γ/IT	^99^Mo/^99m^Tc	[77]
Gallium-67	78.26 h	γ 0.093; 0.185; 0.288; 0.394	EC	^nat^Zn(p,x)^67^Ga ^68^Zn(p,2n)^67^Ga	[78]
Iodine-131	8.02 d	β^−^ 0.606; γ 0.364; 0. 637; 0.284	β^−^	^nat^Te(n,γ)^131^I	[79]
**Therapeutic radionuclides**					
Radium-223	11.43 d	α 5.979	α	^227^Ac/^223^Ra	[80]
Iodine-125	59.4 d	γ 0.035	EC	^124^Xe(n,γ)^125^Xe/^125^I	[81]
Actinium-225	10.0 d	α 5.935	α	^229^Th/^225^Ra/^225^Ac ^226^Ra(p,2n)^225^Ac	[82]

### 2.4. Selection of a Suitable Radiolabelling Strategy for NPs

An ideal radiolabelling strategy for NPs should be easy, fast, robust, highly efficient, and repeatable and must make only minimal changes to the original properties of NPs [83]. The selection of a suitable radiolabelling strategy depends on the chemistry of the radionuclide, the type and surface chemistry of NPs, and/or the final application of radiolabelled NPs. In general, radiolabelling of NPs for imaging and therapy includes two strategies: “direct radiolabelling” and “indirect labelling”. The efficient radiolabelling strategies have been discussed recently by Pellico [84].

“Direct radiolabelling” means the incorporation of radionuclides into a core and/or surface of NPs and may involve the following: (1) mixing radionuclide and non-radioactive nanomaterial precursors during synthesis; (2) chemical adsorption of radionuclide to NPs surface after synthesis, leading to the formation of coordination bonds between chemical groups on NPs surface and radionuclide; (3) incorporation/encapsulation of radionuclide into NPs after synthesis, physical interaction between radionuclide and NPs after synthesis; (4) any mechanism where radionuclide is physically attached to NPs, based on weak electrostatic interactions or driven by the presence of cavities, defects or grooves in the nanomaterial); (5) bombardment of NPs with neutrons or protons, resulting in the transmutation of specific atoms of NPs to others via nuclear reaction).

In the “indirect radiolabelling” approach, the radionuclide is bound to NPs surface by chelators, such as DOTA (1,4,7,10-tetraazacyclododecane-1,4,7,10-tetraacetic acid), NOTA (1,4,7-triazacyclononane-1,4,7-triacetic acid), NODAGA (1,4,7-triazaciclononane, 1-glutaric-4,7-acetic acid), TETA (Triethylenetetramine), etc. [69]. The chelator-based method may involve (1) complexation of a suitable chelator with radiometal and then trapping of radiometal complex during the synthesis of NPs or (2) linking of a suitable chelator to NPs and then complexation of this chelator with radiometal.

## 3. Nanoparticle-Based Radioconjugates for Targeted Prostate Cancer Imaging

Despite the fact that NPs seem to be an attractive platform for the development of tumour-targeted, sensitive, and biocompatible imaging agents, not many NP-based radioconjugates for PCa-targeted imaging have been designed so far (Table 1). They include radiotracers for PET, PET/MRI, PET/NIRF (near-infrared fluorescence imaging), SPECT, SPECT/NIRF, and SPECT/IF (immunofluorescence imaging).

### 3.1. Nanoparticle-Based Radioconjugates for Targeted PET, PET/MR, and PET/NIRF Imaging of PCa

NP-based radioconjugates for PET and PET/MR imaging of PCa have been focused on five positron-emitting radionuclides: ^64^Cu, ^68^Ga, ^18^F, ^124^I, and ^89^Zr. Among them, ^64^Cu was the most frequently used for radiolabelling of NPs due to its attractive nuclear properties [85,86,87]. The characteristics of ^64^Cu and other radionuclides are presented in Table 2.

In 2014, Hu et al. [47] designed and evaluated a multimodal nanoprobe that simultaneously possesses fluorescent, radioactive, and paramagnetic properties. This nanoprobe was designed for in vitro fluorescent imaging and in vivo microPET and MRI imaging of PCa. Nanoprobe was based on 2D tetragonal ultrathin gadolinium vanadate NPs doped with luminescent europium ions (GdVO4:Eu3^+^), radiolabelled with ^64^Cu, and functionalized with Asp-Gly-Ala peptide (DGEA) to target integrin α2β1 (^64^Cu-DOTA-GdVO4:4%Eu-DGEA) (Figure 1).

The cytotoxicity and targeting capability of this nanoprobe were evaluated with PC-3 cells in vitro and further explored in vivo via PET/MR imaging in athymic nude mice (BALB/c nu/nu) bearing PC-3 tumours. The particle distribution in mouse tissues was also determined via fluorescent microscopy. The in vitro experiments showed no significant decrease in cell viability upon incubation of PC-3 cells with nanoprobe. The fluorescent confocal microscopy revealed an intense Eu3+ red signal observed on the surface and in the cytoplasm of PC-3 cells in vitro. The microPET imaging revealed that the DGEA-targeted nanoprobe was five-fold more efficiently uptaken by tumour tissue compared to the untargeted nanoprobe. Apart from the tumour tissue, rapid accumulation of both targeted and untargeted nanoprobes was observed in mouse liver and spleen. The T1-weighted MRI images of the tumour showed slight contrast enhancement in mice 2 and 4 h post-injection of targeted and untargeted nanoprobes; however, 24 h post-injection, the MRI images of the tumour showed significant contrast enhancement in mice exposed to the nanoprobe targeted with DGEA ligand compared to untargeted nanoprobe.

Wong et al. (2017) [42] proposed a novel probe for PET imaging, based on lipid micelles modified with polyethylene glycol (PEG), functionalized with anti-PSMA single chain antibody (scFv) fragment, conjugated with the metal chelate (DOTA), and radiolabelled with ^64^Cu (Figure 2).

The authors tested the usefulness of two different thiol ester-PEGs for attachment of the single chain antibody (scFv) fragment, namely DSPE-PEG2000-maleimide (mal) and DSPE-PEG2000-acetamidobromomide (acetBr). Final compounds were evaluated in NOD/SCID mice bearing human prostate cancer LNCap tumours. The ^64^Cu-PET imaging and distribution of radioactivity in mouse tissue showed that both ^64^Cu-DOTA-mal-scFv-cys-LNP and ^64^Cu-DOTA-acetBr-scFv-cys-LNP conjugates were two-fold more efficiently uptaken by the tumour tissue than NP-free PSMA-targeted conjugates ^64^Cu-DOTA-mal-scFv-cys and ^64^Cu-DOTA-acetBr-scFv-cys. The PSMA-targeted NPs showed a 60% increase in tumour uptake compared to the non-targeted NPs. For the PSMA-targeted conjugates, accumulation of radioactivity was observed in the liver, kidney, spleen, bladder, and tumour over 2 days. The untargeted conjugates were predominantly uptaken by the liver and showed slow clearance.

In 2018, Cai et al. [45] presented a unique direct targeting approach based on bio-conjugation of bombesin peptide to copper sulphide NPs with ^64^Cu radionuclide integrated into the CuS core ([^64^Cu]CuS) (Figure 3). The authors designed two kinds of NP-based radioconjugates, targeted Bom-PEG-[^64^Cu]CuS radioconjugate and untargeted PEG-[^64^Cu]CuS) radioconjugate, and determined their specific binding and cellular uptake in vitro into aggressive prostate cancer cells PC-3-KD1. The ability of these radioconjugates to accumulate in orthotopic prostate tumours was further evaluated via PET imaging in nu/nu mice bearing PC-3-KD1 tumours. The in vitro results showed that bombesin-targeted showed a much faster cellular binding rate and far better uptake efficiency compared with untargeted NPs.

The PET/CT imaging revealed that in vivo uptake of untargeted NPs by tumour tissue was very low and ranged from 1.2 and 1.4% (ID/g) after 1 and 6 h, respectively. In contrast, the bombesin-targeted radioconjugate gradually accumulated in the tumour tissue, displaying enhanced tumour-to-surrounding tissue contrast and significantly higher tumour accumulation levels (3.5 and 5.0% (ID/g after 1 and 6 h, respectively). Analysis of biodistribution confirmed the PET/CT data and showed that both bombesin-targeted and -untargeted NPs were highly uptaken by liver. Moreover, untargeted NPs displayed significantly higher uptake than bombesin-targeted NPs in the spleen, suggesting the active targeting capability of the Bom-PEG-[^64^Cu]CuS NPs had the potential to reduce a nonspecific uptake of NPs by the mononuclear phagocytic system.

The ^68^Ga-labelled NPs for targeted PET and PET/MRI imaging in PCa were used less often than ^64^Cu-labelled ones. In 2016, Moon et al. [47] developed a dual-modal PET/MRI imaging probe, ^68^Ga-DOTA-IO-GUL, composed of iron oxide (IO) nanoparticles, modified with polysorbate 60 and PEG/DOTA chains radiolabelled with ^68^Ga, and functionalized with anti-PSMA glutamate-ureid-lysine (GUL) moieties (Figure 4).

The in vitro competitive binding study showed a dose-dependent binding of DOTA-IO-GUL to PSMA-positive LNCaP cells. Moreover, the PET and MRI results revealed selective uptake by PSMA-positive 22Rv1 cells but not by PSMA-negative PC3 cells in BALB/c nude mice bearing xenografts of 22Rv1 or PC3 tumours. Though the resolution of single-modal MR images was higher than the single-modal PET images, the quantitative information provided was still limited. Unlike the single-modal PET images that provided quantitative information, the resolution of PET images was lower than the MR images. However, a dual-modal PET/MRI conjugate ^68^Ga-DOTA-IO-GUL possessed the complementary effect of using both MR and PET imaging.

More recently, a bispecific iron oxide NPs ^68^Ga-mNP-N1/2 was synthesized for PET/MR dual-modality imaging of PCa tumours overexpressing PSMA or/and GRPR [48]. These magnetic iron oxide NPs were covered with a thin silica layer carrying -SH (mNP-S1/2) or –NH2 groups (mNP-N1/2). The modified NPs were functionalized with Glu-ureido-based PSMA ligand, targeting PSMA, and with bombesin peptide, targeting GRPR. The resulting heterobivalent NPs were radiolabelled with ^68^Ga using a direct labelling procedure (Figure 5).

The ^68^Ga-mNP-N1/2 conjugate proved superior to the ^68^Ga-mNP-S1/2 conjugate regarding radiolabelling efficiency and was further evaluated in vitro. The aim of combining two ligands targeting receptors of different densities on well and poorly differentiated prostate cancers was to increase the efficiency of PET/MR imaging in the case of tumour heterogeneity. Toxicity was studied in vitro in LNCaP (PSMA-positive and GRPR-negative) and PC-3 (PSMA-negative and GRPR-positive) cells. In vitro results showed specific time-dependent binding (40 min to plateau), high avidity (PC-3: Kd = 28.27 nM, LNCaP: Kd = 11.49 nM) and high internalization rates for ^68^Ga-mNP-N1/2 in both cell lines. The in vitro hemolysis assay results showed low toxicity of ^68^Ga-mNP-N1/2 conjugate.

An interesting approach for dual-targeting and dual-modality PET/NIRF imaging was proposed by Hu et al. [49]. The authors developed amine-modified cadmium telluride quantum dots (CdTeQDs) with symmetric β-glutamate (β-Glu) and tripolyethylene glycol (PEG3) groups, radiolabelled with ^1^8F and functionalized with RGD-peptide targeting integrin αvβ3 and with bombesin peptide (7–14) (BBN) targeting GRPR (^18^F-FP-CdTeQD-RGD-BBN) (Figure 6).

Cytotoxicity in vitro and cell-binding studies were performed with prostate cancer PC-3 cells. In vivo dual-modality PET/NIRF imaging of ^18^F-FP-CdTeQDs-RGD-BBN was performed in nu/nu BALB/c nude mice bearing PC-3 tumours. The NIRF imaging demonstrated a weak fluorescent signal 30 min after injection and steadily increasing fluorescence in tumours within several hours after injection. The PET imaging revealed radioactivity accumulation in the tumours with a peak at 60 min after injection. The radioactivity accumulation in the brain and muscle decreased from 30 to 120 min after injection.

Xia et al. [44] developed melanin NPs, modified with PEG5000 chains, functionalized with PSMA-SH small molecule inhibitor, and directly radiolabelled with ^124^I (Figure 7).

In vitro studies proved high specificity, efficient cellular uptake, and biocompatibility of radioconjugate ^124^I-PPMN in LNCaP and 22RV1 prostate cancer cells. The PET imaging showed significantly higher uptake by tumour tissue in mice bearing LNCaP tumours compared with mice bearing 22RV1 tumours for up to 72 h. The biodistribution results revealed that radioactivity of ^124^I-PPMN was mainly accumulated in the heart, liver, spleen, intestine, and tumour. Long retention of the ^124^I-PPMN in the tumour offered imaging time flexibly and allowed for long-term monitoring of the therapeutic effect and potentiated therapeutic effect of the radionuclide.

In 2022, Meher et al. [40] designed and synthesized three 4-armed starPEG40kDa nanocarriers, functionalized with zero, one, or three molecules of urea-bearing PSMA-targeted (ACUPA) ligands and radiolabelled with ^89^Zr via desferrioxamine B (DFO-B) chelator (Figure 8). In vitro cellular uptake and internalization of [^89^Zr]PEG-(DFB)3, [^89^Zr]PEG-(DFB)3(ACUPA)1 and [^89^Zr]PEG-(DFB)1(ACUPA)3 were tested in PSMA-positive PC3-Pip and PSMA-negative PC3-Flu cells. In vivo PET imaging of the ^89^Zr-labelled nanocarriers and organ biodistribution studies were performed in the nu/nu athymic mice bearing subcutaneous dual xenografts of PC3-Pip and PC3-Flu tumours. In vitro results revealed that the use of multivalent PSMA binder [^89^Zr]PEG-(DFB)1(ACUPA)3 with three copies of ACUPA ligands increased PSMA binding affinity and cellular internalization compared with the conjugate containing the single ACUPA ligand.

In vivo and ex vivo studies revealed that both conjugates demonstrated remarkably higher tumour retention and background clearance in PSMA-positive PC3-Pip tumours compared to nontargeted conjugates. Although targeting significantly improved tumour retention and tissue penetration of both nanocarriers in PSMA-positive PC3-Pip xenografts, the multivalent nanocarrier [^89^Zr]PEG-(DFB)1(ACUPA)3 bearing three ACUPA ligands showed a remarkably higher PC3-Pip/blood ratio and background clearance.

### 3.2. NP-Based Radioconjugates for PCa Targeted SPECT, SPECT/IF, and SPECT/NIRF Imaging

The NP-based radioconjugates for SPECT, SPECT/IF, and SPECT/NIRF imaging of PCa have been focused on three γ-emitting radionuclides: indium-111, metastable technetium-99m, and gallium-67. The characteristics of these radionuclides are presented in Table 2.

Zhang et al. [41] designed and evaluated core-crosslinked polymeric micellar NPs (CCPM) for dual, SPECT and optical, imaging of PCa. The NPs were modified with PEG, functionalized with TNYL-RAW peptide targeting the EphB4 receptor, and then labelled with ^111^In and with a NIR fluorescent indocyanine 7 (Cy7)-like dye. In this study, Cy-7-like dye was entrapped in the core of CCPM, whereas ^111^In was conjugated with the surface of CCPM by a DTPA chelator (Figure 9).

In vitro studies showed that TNYL-RAW-CCPM NPs selectively bound to EphB4-positive PC-3M prostate cancer cells but not to EphB4-negative A549 lung cancer cells. The pharmacokinetic data indicated a higher and longer accumulation of NP-free ^111^In-TNYL-RAW conjugate in blood compared with the NP-based ^111^In-TNYL-RAW conjugate. The complementary results acquired with SPECT and NIRF in mice bearing the EphB4-positive PC-3 tumour revealed two times higher uptake by the tumour tissue and tumour-to-blood ratio for the NP-based conjugate ^111^In-TNYL-RAW-CCPM than for the NP-free 111In-TNYL-RAW conjugate. The biodistribution study confirmed the SPECT and NIRF results. Moreover, immunohistochemical analysis showed that NIRF signal from NPs correlated with their radioactivity count and co-localized with the EphB4 expressing region.

Banerjee et al. [39] synthesized and tested the PSMA-targeted poly(lactic acid)-polyethyene glycol (PLA–PEG) copolymer-based NPs for in vivo SPECT/CT and ex vivo NIRF imaging. These NPs were modified with terminal PEG groups, functionalized with ACUPA, radiolabelled with ^111^In by DOTA chelation, or labelled with IRDye 680RD infrared dye (Figure 10).

The tissue biodistribution studies in athymic NOD SCID nude mice bearing PSMA-positive PC3 PIP and PSMA-negative PC3 flu tumours revealed similar accumulation of the radioactive PSMA-targeted and untargeted NPs in all tissues, except for the tumour tissue and liver. The tumour and liver retention of the PSMA-targeted NPs was slightly longer than untargeted ones. However, the clearance of untargeted NPs from the PSMA-positive PC-3 PIP tumour was significantly faster compared to the PSMA-targeted NPs. SPECT/CT imaging confirmed the biodistribution results and demonstrated higher tumour uptake of the radioactive PSMA-targeted NPs compared to the untargeted NPs. To perform ex vivo biodistribution and microscopy studies, the authors labelled the PSMA-targeted and untargeted NPs with a NIRF probe using IRDye 680RD infrared dye. The results revealed that both the PSMA-targeted and untargeted NPs accumulated within tumours. However, the PSMA-targeted NPs were uptaken by the PSMA-positive tumour epithelial cells and tumour-associated macrophages, while the untargeted NPs were primarily uptaken by macrophages.

Mendoza-Sánchez et al. [31] developed multimodal gold NPs (AuNPs), functionalized with Lys3-bombesin peptide targeting GRPR and modified with HYNIC-Gly-Gly-Cys-NH2 (HYNIC-GGC) peptide, radiolabelled with ^99m^Tc ([^99m^Tc]-AuNP-Lys3-bombesin) for in vivo SPECT imaging (Figure 11).

In vitro binding studies revealed that the NPs are able to specifically recognise the GRP receptors overexpressed in prostate cancer PC-3 cells. Cellular uptake of the targeted NPs ([^99m^Tc]-AuNP-Lys3-bombesin) was significantly higher than the untargeted NPs (99mTc-AuNP). Biodistribution studies and in vivo micro-SPECT/CT images of athymic mice bearing PC-3 tumours showed an evident accumulation of [^99m^Tc]-AuNP-Lys3-bombesin conjugate in tumour, liver, and spleen. The results revealed a 1.5 times higher uptake by the tumour tissue and tumour-to-blood ratio for the targeted NP-based conjugate than for targeted NP-free [^99m^Tc]-HYNIC-Lys3-bombesin conjugate.

Felber et al. [33] synthesized AuNPs and CdSe/ZnS core-shell QDs, which were modified with bifunctional ligands (HS-PEG-DAP and HS-PEG-DAP-TF) then radiolabelled with ^99m^Tc and functionalized with a small molecule urea-based PSMA-I inhibitor (Figure 12).

In vitro studies showed high stability of radiolabelled AuNPs, whereas for QDs, partial detachment of the coating ligands was observed. In vitro studies showed active uptake of AuNPs by LNCaP cells. Unfortunately, in vivo imaging with microSPECT of nude NMRI mice bearing LNCaP xenografts and ex vivo biodistribution studies showed low accumulation of radiolabelled AuNPs in the tumour tissue. Moreover, these studies showed rapid clearance of the NPs by the spleen and liver resulting in a short time of circulation in the blood.

To date, only one study with ^67^Ga-based radiolabelled NPs was presented for targeted SPECT imaging of PCa. Zambre et al. [32] developed AuNPs modified with DTPA (diethylene triamine penta-acetic acid) linked to a surface of AuNPs via dithiol (DT) linkage, functionalized with bombesin peptide (BBN), and radiolabelled with 67Ga (Figure 13).

Biodistribution studies of this radioconjugate [BBN-AuNP(DTDTPA)(^67^Ga)] in nude mice bearing PC-3 tumour showed ~six-fold higher tumour uptake after 24 h post-injection compared with non-targeted AuNP-DTDTPA-67Ga.

### 3.3. NP-Based Radioconjugates for PCa-Targeted Therapy

The NP-based radioconjugates for PCa-targeted therapy have been focused on two α-emitting radionuclides: actinium-225 and radium-223. The characteristics of these radionuclides are presented in Table 2.

In 2015, Bandekar and co-workers designed and evaluated PEGylated liposomal NPs, which were loaded with ^225^Ac and functionalized with mouse anti-human PSMA J591 antibody or with A10 PSMA aptamer for the PSMA-targeted radiotherapy of PCa (Figure 14) [34]. Liposomal NPs were composed of 1,2-dinonadecanoyl-sn-glycero-3-phosphocholine (DNPC):cholesterol:1,2-distearoyl-sn-glycero-3-phosphoethanol-amine-N-[methoxy(polyethyleneglycol)-2000](ammonium salt) (DSPE-PEG):1,2-dipalmitoyl-sn-glycero-3-phosphoethanolamine-N-(lissamine rhodamine B sulfonyl) (ammonium salt) (DPPE-rhodamine) and were prepared to encapsulate DOTA chelator. 225Ac was loaded into preformed liposomes. In vitro biological studies showed that the NP-based radioconjugate with J591 antibody and the NP-free radioconjugate with J591 antibody exhibited similar cytotoxicity.

Both radioconjugates were similarly internalised, ranging between 25 and 36%. However, in vitro uptake studies on LNCaP and Mat-Lu prostatic cells and on human umbilical vein endothelial cells (HUVEC) artificially expressing PSMA demonstrated increased cellular binding, internalization, and radiotoxicity of the NP-based radioconjugate targeted with J591 antibody compared to the NP-based radioconjugate with A10 PSMA aptamer or to the untargeted liposomal NPs.

Other lipid-based NPs were designed by Zhu et al. [35] and comprised 21PC (2-diheneicosanoyl-sn-glycero-3-phosphocholine): Cholesterol: DSPE-PEG(1,2-distearoyl-sn-glycero-3-phosphoethanolamine-N-[methoxy(PEG2000)]: DPPE-rhodamine (1,2)-dipalmitoyl-sn-glycero-3-phosphoethanol amine-N-(lissaminerhodamine B Sulfonyl), loaded with ^225^Ac and functionalized with a fully human anti-PSMA J591 antibody or with a low-molecular-weight urea-based PSMA inhibitor (Figure 15). In vitro studies performed on PSMA-positive and PSMA-negative HUVEC cells revealed that NP-based radioconjugates functionalized with a fully human anti-PSMA J591 antibody or with a low-molecular-weight urea-based PSMA inhibitor exhibited similar killing efficacy, which was increased almost three-fold compared to the cell killing efficacy of the NP-free PSMA-targeting antibody. The increase in killing efficacy was accompanied by elevated levels of DNA double-strand breaks and strongly correlated with intracellular uptake. Both types of NP-based conjugates exhibited nucleo-cytoplasmic localization unlike the NP-free PSMA-targeting antibody, which preferentially localised near the plasma membrane.

Recently, a novel NP-based radioconjugate [^223^Ra]A-silane-PEG-D2B for targeted alpha therapy of PCa was successfully synthesised and characterised [50,59]. This compound consisted of a NaA zeolite NPs loaded with 223Ra, modified with silane-PEG molecules, and functionalized with anti-PSMA D2B antibody (Figure 16).

The competition binding studies revealed the high affinity of this NP-based radioconjugate towards the PSMA receptor and its very fast and selective internalization into PSMA-positive LNCaP C4-2 cells but not into PSMA-negative DU-145 cells. The analysis of cytotoxicity confirmed that this conjugate was about four-fold more toxic for LNCaP C4-2 cells than for DU-145 cells. Biodistribution studies in BALB/c nude mice bearing LNCaP C4-2 tumour revealed a high accumulation of the NP-based radioconjugate in the liver, lungs, spleen, and bone tissue. Unfortunately, both the PSMA-targeted and untargeted NP-based radioconjugates exhibited a similar marginal uptake in tumour tissue, indicating that intravenous administration of the radioconjugate is dubious due to the lack of effective delivery to the tumour tissue.

### 3.4. NP-Based Radioconjugates for Theranostic Applications in PCa

Theranostic NPs are multifunctional nanosystems for specific and targeted disease management due to incorporating desirable diagnostic, imaging, and therapeutic capabilities into one single nanoparticle. The combination of multiple functionalities enables a variety of integrated imaging and treatment protocols and allows for the reduction in adverse effects on normal tissue [88,89]. Although significant progress has been made during the last decade in developing theranostic NPs for cancer imaging and therapy, they are still in a very early translational stage [89]. To date, only five examples of NP-based radioconjugates for theranostic applications in PCa have been reported.

Yallapu et al. [38] evaluated the theranostic potential of poly(lactic-co-glycolic acid) (PL-GA) NPs loaded with polyphenol curcumin, functionalized with the anti-PSMA J591 antibody, and radiolabelled with ^131^I, which simultaneously emits β^−^ radiation useful for the treatment and γ radiation useful for diagnosis ([^131^I]-PSMA-PLGA-CUR) (Figure 17).

The results indicated that these NPs efficiently inhibited the growth of PCa cells both in vitro and in vivo. The NP-based radioconjugate [^131^I]-PSMA-PLGA-CUR accumulated specifically in the tumour tissue in a dose-dependent manner in the LNCaP C4-2 xenograft mice model. The whole-body and ex vivo imaging of multiple organs proved that the NP-based radioconjugate retained within the tumour tissue at a much higher level than the NP-free [^131^I]-PSMA targeting antibody. In contrast to the tumour, other organs exhibited minimal 131I radioactivity from the NP-based radioconjugate.

Wang et al. [30] prepared dual-labelled theranostic silicon NPs for SPECT imaging in vivo, allowing also for tissue-level localization ex vivo by means of fluorescence microscopy. The undecylenic acid-modified thermally hydrocarbonized porous silicon nanoparticles (UnTHCPSi) were modified with the dibenzo cyclooctyne (DBCO) linker, functionalized with iRGD peptide targeting the αVβ3 receptor, labelled with a fluorescent Alexa Fluor 488 dye, and radiolabelled with ^111^In by DOTA chelation. The hydrophobic antiangiogenic drug sorafenib was loaded onto these nanoparticles for chemotherapeutic applications (Figure 18).

The SPECT imaging studies one hour after intravenous injection of the radioconjugate to Hsd:NMRI-Foxnl nu/nu nude mice bearing PC3-MM2 tumours revealed that both the untargeted 111In-PSi NPs and the αVβ3 receptor targeted [^111^In]-PSi-iRGD NPs accumulated mainly in liver and spleen, with minor radioactivity distributed in other organs and no radioactivity visible in tumour tissue. However, long-term tissue biodistribution studies showed that the αVβ3 receptor targeted [^111^In]-PSi-iRGD NPs displayed higher tumour-specific accumulation compared to the untargeted [^111^In]PSi NPs. After administration directly to the tumour, the αVβ3 receptor targeted [^111^In]-PSi-iRGD NPs showed higher tumour retention compared to the untargeted [^111^In]-Psi NPs. This was translated to efficient suppression of PC3-MM2 prostate cancer xenograft growth when sorafenib-containing NPs were delivered directly to the tumour. The higher tumour uptake of the αVβ3 receptor targeted [^111^In]-PSi-iRGD NPs was confirmed via the Alexa Fluor 488 immunofluorescence staining.

Wong et al. [43] synthesized and evaluated imaging and therapeutic potential of homogenous covalent mixture of doxorubicin-attached PEG micelles radiolabelled with ^64^Cu [^64^Cu]DOTA-Dox-PEG-LNP], PEG micelles functionalized with anti-PSMA diabody (cys-DB) and radiolabelled with ^64^Cu [^64^Cu]DOTA-cys-DB-PEG-LNP, and NP-free doxorubicin radiolabelled with ^64^Cu [^64^Cu]DOTA-Dox (Figure 19).

A pharmacokinetic study revealed that blood clearance of a covalent mixture of NP-based [^64^Cu]DOTA-cys-DB-PEG-LNP and [^64^Cu]DOTA-Dox-PEG-LNP was slower than NP-free [^64^Cu]DOTA-Dox or NP-based [^64^Cu]DOTA-Dox-PEG-LNP alone. PET imaging of NOD/SCID mice bearing LnCaP tumours showed superior tumour targeting by the mixture of [^64^Cu]DOTA-cys-DB-PEG-LNP and [^64^Cu]DOTA-Dox-PEG-LNP (15% ID/g) compared to the NP-based untargeted [^64^Cu]DOTA-Dox-PEG-LNP radioconjugate (10% ID/g) and NP-free untargeted [^64^Cu]DOTA-Dox radioconjugate (3% ID/g). Moreover, a four-fold increase in tumour uptake was observed for NP-based untargeted radioconjugate containing doxorubicin [^64^Cu]Dox-DOTA-PEG-LNP (10% ID/g) compared to NP-free untargeted doxorubicin [^64^Cu]-DOTA-Dox (3% ID/g).

An active targeting probe was reported by Yari et al. [36], who designed the P3 lipopolymer that could be used to functionalize liposomes for targeted delivery of therapeutics/diagnostics to PSMA-positive PCa cells. Liposomes were modified by post-insertion of a lipopolymer (P^3^), comprising a small molecule Lys–urea–Glu-based PSMA inhibitor (PSMAL), PEG2000, and palmitate linker, and radiolabelled with ^99m^Tc radionuclide or loaded with doxorubicin (Figure 20).

In vitro cellular uptake and toxicity of [^99m^Tc]P3-liposomes were determined in PSMA-positive LNCaP and PSMA-negative PC3 cells and compared with [^99m^Tc]plain-liposomes without the PSMAL inhibitor. In vitro research revealed circa three-fold higher radioactivity in LNCaP cells treated with [^99m^Tc]P3-liposomes compared to the cells treated with [^99m^Tc]plain-liposomes. These authors investigated also whether P3-liposomes are able to deliver cytotoxic concentrations of doxorubicin to LNCaP and PC3 cells. The cytotoxicity assay results showed that doxorubicin-loaded P^3^-liposomes with PSMAL were significantly more toxic to PSMA-positive LNCaP cells compared to PSMA-negative PC3 cells. Moreover, doxorubicin-P3-liposomes were more cytotoxic than doxorubicin-plain-liposomes in LNCaP cells. At the same time, there was no difference in the cytotoxicity profile between doxorubicin-P3-liposomes and doxorubicin-plain-liposomes in PC3 cells.

Recently, Cheng et al. [37] developed a “one-for-all” approach for assembling theranostic texaphyrin NPs for PSMA-targeted radionuclide imaging and focal photodynamic therapy (PDT). These texaphyrin−phospholipid NPs were labelled with ^111^In and ^175^Lu isotopes and functionalized with urea-bearing PSMA-targeting YC-XII-35 moiety (Figure 21).

Biodistribution studies of these NPs by NIR fluorescence, SPECT/CT imaging, and γ counting revealed a significant uptake of the PSMA-targeted [^111^In/^175^Lu]-texaphyrin NPs in tumour tissue of mice bearing the PSMA-positive PC3 PIP tumour compared to mice bearing the PSMA-negative PC3 flu tumour. Moreover, the PSMA-targeted [^111^In/^175^Lu]-texaphyrin NPs when illuminated with light showed a potent PDT effect and successfully inhibited PSMA-positive PC3 PIP tumour growth in a subcutaneous xenograft model.

### 3.5. Clinical Studies on NP-Based Radioconjugates for Targeted Imaging and Therapy of PCa

So far, only one clinical trial has addressed the NP-based radioconjugates for targeted PCa imaging: NCT04167969 (November 2019–November 2023). The purpose of this study entitled, “Molecular Phenotyping and Image-Guided Surgical Treatment of Prostate Cancer Using Ultrasmall Silica Nanoparticles”, is to determine the usefulness of [^64^Cu]-NOTA-PSMA-PEG-Cy5.5-C′ tracer as a safe and reliable tool in identifying tumour cells before and during surgery and to find whether PET/MRI scans conducted after injection of this NP-based tracer are more accurate than the usual imaging scans used to locate deposits of prostate tumour cells.

## 4. Conclusions

Recent developments in the field of NP-based radioconjugates for targeted imaging and therapy of PCa have revealed a wide diversity of design approaches and methods of preparation. The most important advantage of these radioconjugates is highly specific targeting (Appendix A). This property enabled hitting the PCa effectively and with high precision. The next important advantage of all these radioconjugates is the high stability of the radionuclide–nanoparticle binding, which allowed for analysis of the real distribution of NP-based radioconjugate and imaging reliability. Another advantage of some NP-based radioconjugates is their multimodality. The combination of imaging modalities provided simultaneous functional and morphological information and overcame the limitations of the independent techniques. The next important advantage of these radioconjugates is very low or neglectable toxicity in vivo, resulting both from the biocompatibility of the vast majority of the nanoparticles used as well as from the fast clearance of radionuclides by the kidneys, liver, and urinary bladder. For therapeutic and theranostic applications, these NP-based radioconjugates have several advantages over targeting ligand radioconjugates. First, nanoparticles allow encapsulation of the parent radionuclide inside (e.g., ^225^Ac and ^223^Ra). This prevents daughter radionuclides from escaping from the target site and reduces toxic side effects. In addition, nanoparticles allow overcoming difficulty in the stable attachment of some radionuclides (e.g., ^223^Ra) to the targeting ligands. Moreover, they allow encapsulation and delivery of drugs (e.g., doxorubicin, monomethyl auristatin E (MMAE), sorafenib, and curcumin) into the tumour. The in vivo results confirmed the controlled release of the drug over a long period of time, increasing therapeutic index, accumulation into the tumour tissues, and inhibition of the tumour growth. Furthermore, the most important advantage of some NP-based radioconjugates is the possibility to deliver the drug and imaging agent(s) at the targeted site to diagnose and treat the PCa. Despite these many advantages of the NP-based radioconjugates, there are still some disadvantages/challenges hampering their translation to the clinic. One of the biggest problems in the use of NP-based radioconjugates is their entrapment in mononuclear phagocytic systems. Surface modification of NPs with polyethylene glycol (PEG) prevents their agglomeration and results in prolonged presence in circulation due to hindered recognition and phagocytosis, but this strategy is not efficient against accumulation in non-tumour tissues. The next important disadvantage of these radioconjugates is the lack of adequate knowledge about their long-term impact on cellular signalling pathways and biochemical processes of the human body. Another disadvantage is poor control of degradation, pharmacokinetics, and biodistribution in vivo. Therefore, for each promising NP-based radioconjugate evaluated in preclinical studies in vitro, the risk-to-benefit ratio should be determined in vivo before introducing them to the clinic.

## Figures and Tables

**Figure 1 molecules-28-04122-f001:**
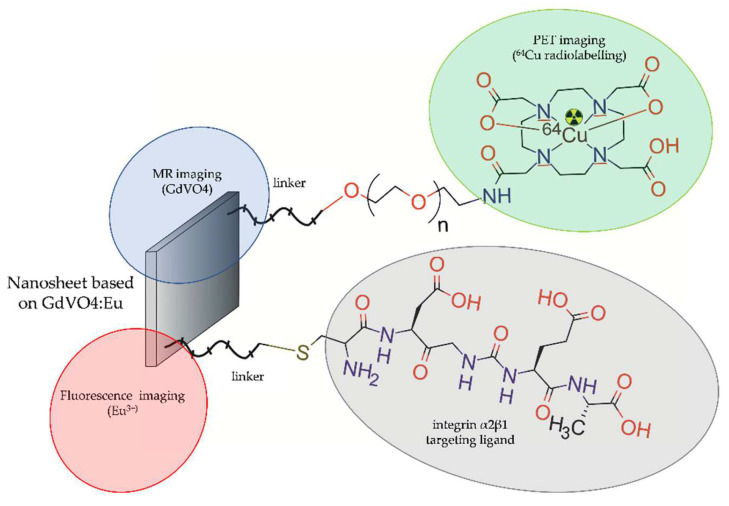
Structure of [^64^Cu]-DOTA-GdVO4:4%Eu-DGEA radioconjugate consisting of GdVO4:Eu ultrathin nanosheet functionalized with Asp-Gly-Ala peptide and radiolabelled with ^64^Cu.

**Figure 2 molecules-28-04122-f002:**
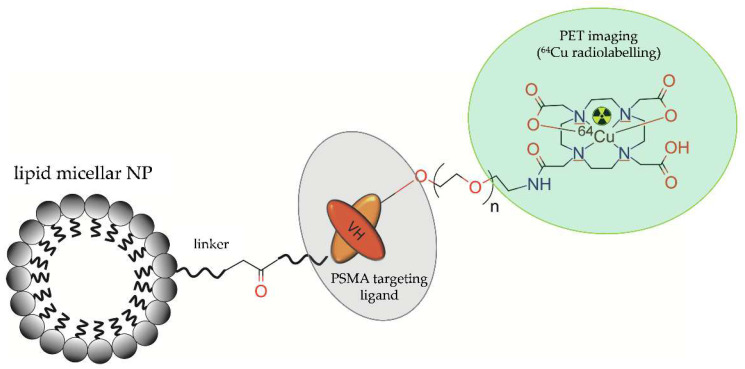
Structure of [^64^Cu]-DOTA-acetBr-scFv-cys-LNP radioconjugate consisting of lipid micellar NP functionalized with a single chain antibody (scFv) fragment and radiolabelled with ^64^Cu.

**Figure 3 molecules-28-04122-f003:**
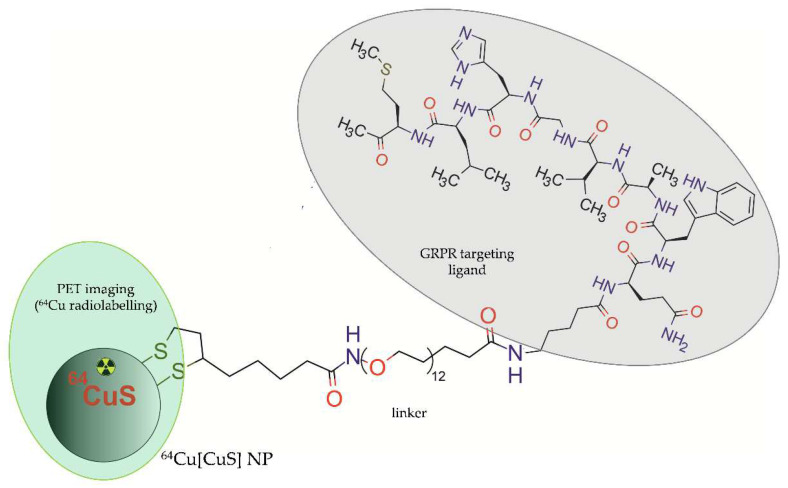
Structure of Bom-PEG-[^64^Cu]CuS radioconjugate consisting of copper sulphide NP with integrated ^64^Cu and functionalized with bombesin peptide.

**Figure 4 molecules-28-04122-f004:**
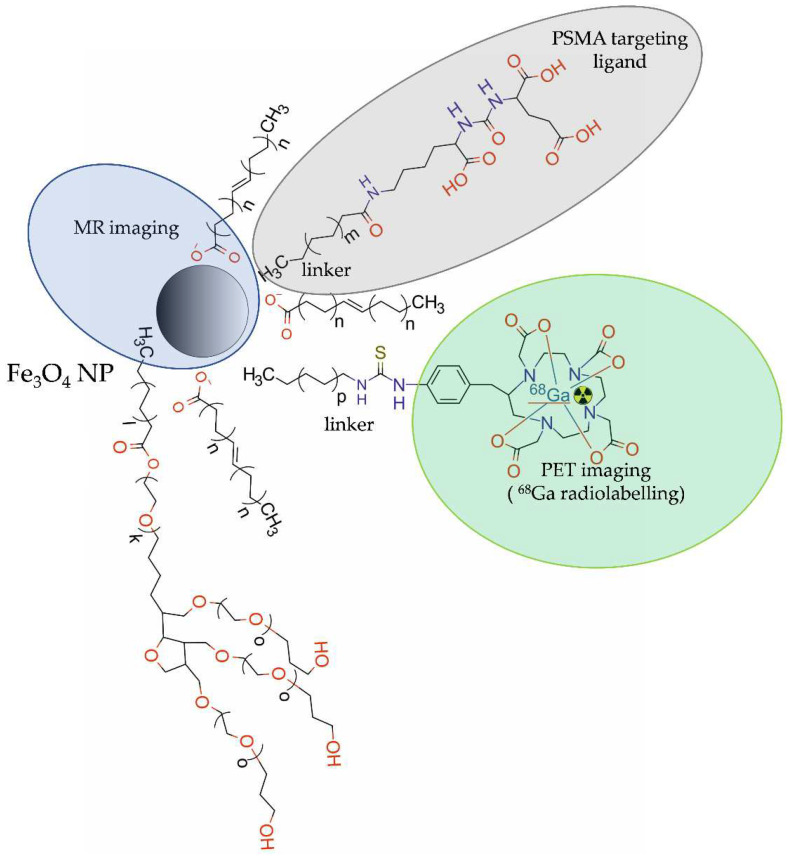
Structure of [^68^Ga]-DOTA-IO-GUL radioconjugate consisting of iron oxide NP modified with PEG chains and DOTA, functionalized with glutamate-ureido-lysine (GUL) moieties, and radiolabelled with ^68^Ga.

**Figure 5 molecules-28-04122-f005:**
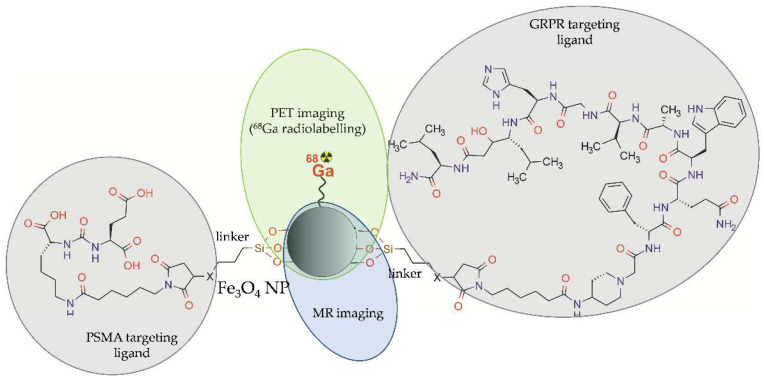
Structure of [^68^Ga]-mNP-N1/2 radioconjugate consisting of iron oxide NP functionalized with Lys-ureido-Glu moiety (1) and bombesin (2) and radiolabelled with ^68^Ga.

**Figure 6 molecules-28-04122-f006:**
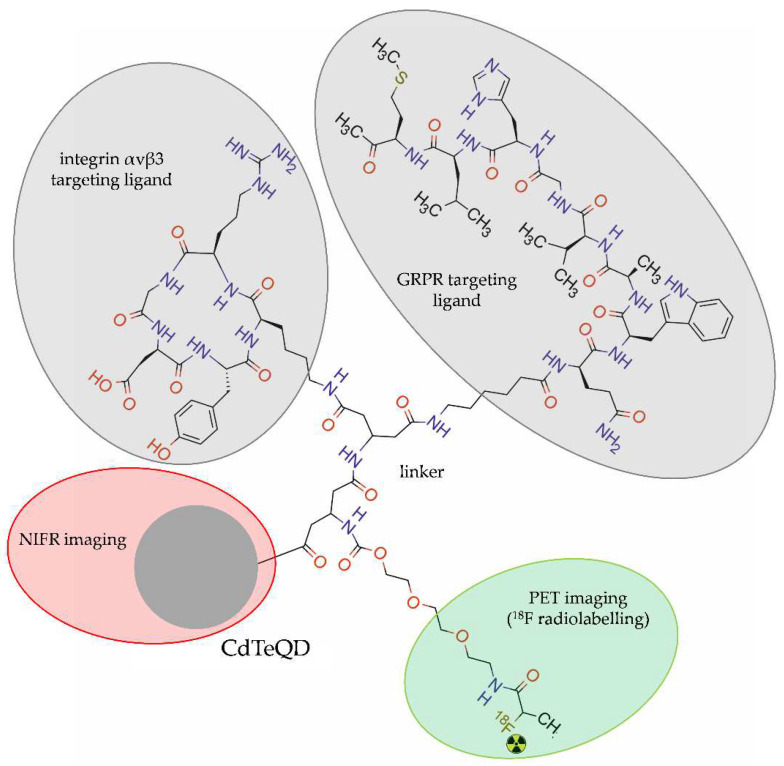
Structure of [^18^F]-FP-QDs-RGD-BBN radioconjugate consisting of cadmium telluride quantum dot (CdTeQDs), modified with β-glutamate group and tripolyethylene glycol, double functionalized with arginine-glycine-aspartate acid (RGD) and bombesin (7–14), and radiolabelled with ^18^F.

**Figure 7 molecules-28-04122-f007:**
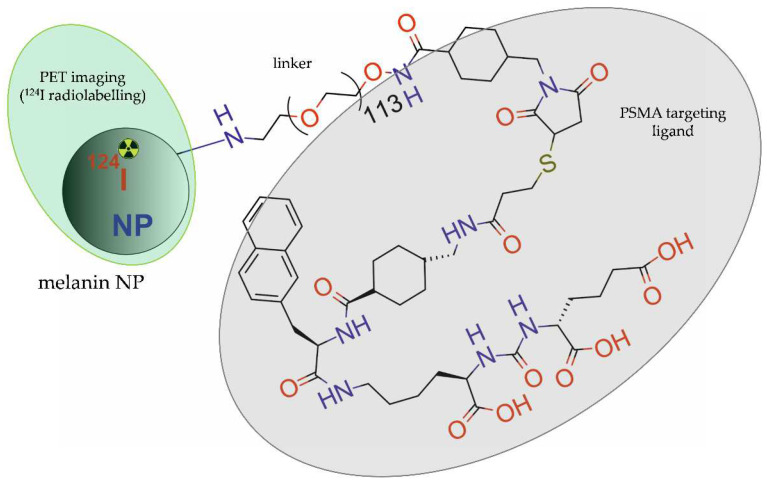
Structure of [^124^I]-PPMN radioconjugate consisting of melanin NP modified with PEG_5000_ chains, functionalized with PSMA-SH small molecule inhibitor, and radiolabelled with ^124^I.

**Figure 8 molecules-28-04122-f008:**
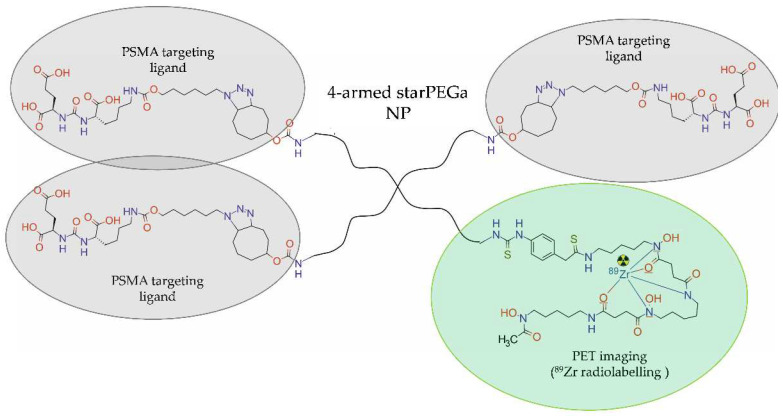
Structure of [^89^Zr]PEG-(DFB)1(ACUPA)3 radioconjugate consisting of four-armed starPEG40kDa NP functionalized with the urea-bearing PSMA-targeted (ACUPA) ligands and radiolabelled with ^89^Zr.

**Figure 9 molecules-28-04122-f009:**
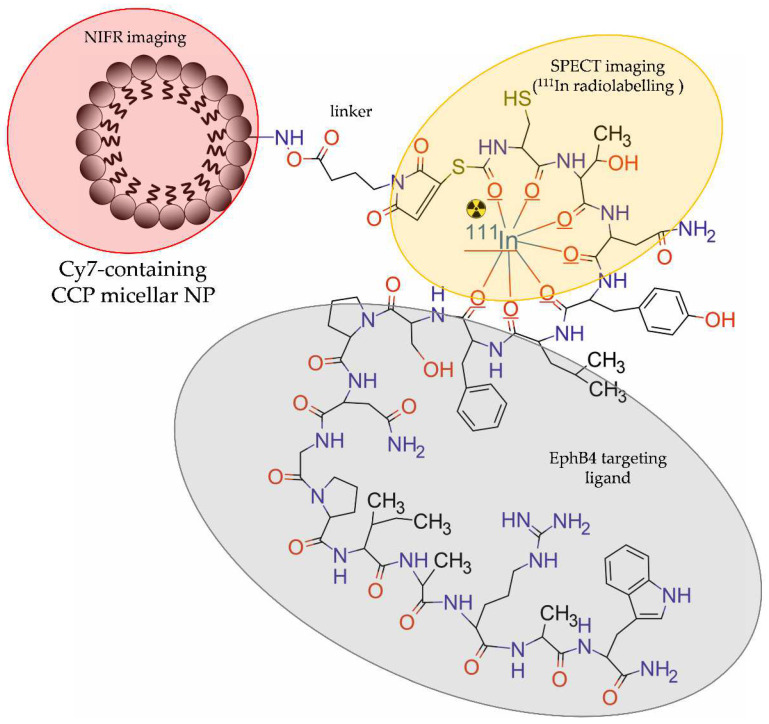
Structure of [^111^In]-TNYL-RAW-CCPM radioconjugate consisting of core-crosslinked polymeric micellar NP modified with PEG, functionalized with TNYL-RAW peptide, labelled with Cy7-like dye, and radiolabelled with ^111^In.

**Figure 10 molecules-28-04122-f010:**
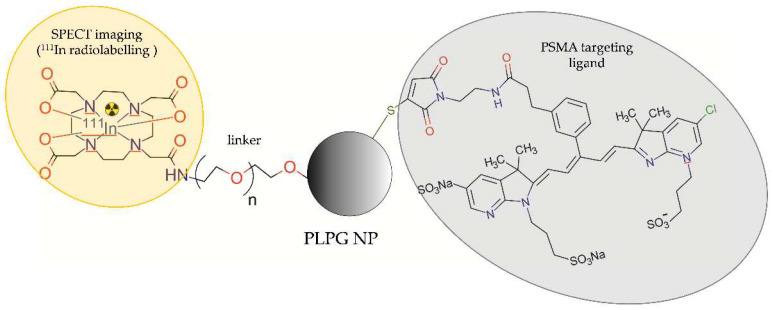
Structure of [^111^In]DOTA-PEG-PLA–PEG-ACUPA radioconjugate consisting of poly(lactic acid)-polyethyene glycol copolymer-based NP functionalized with a urea-bearing moiety (ACUPA) and radiolabelled with ^111^In.

**Figure 11 molecules-28-04122-f011:**
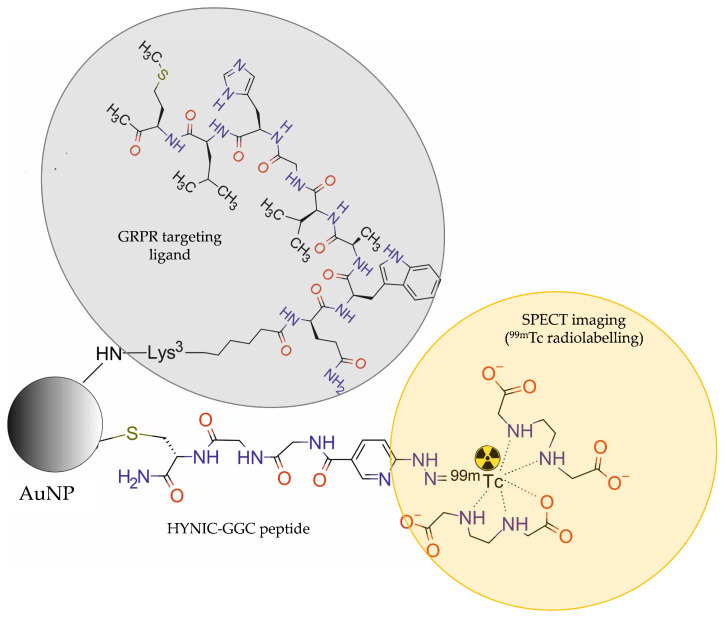
Structure of [^99m^Tc]AuNP-Lys3-bombesin radioconjugate consisting of gold NP modified with HYNIC-Gly-Gly-Cys-NH2 (HYNIC-GGC) peptide, functionalized with Lys3-bombesin, and labelled with ^99m^Tc.

**Figure 12 molecules-28-04122-f012:**
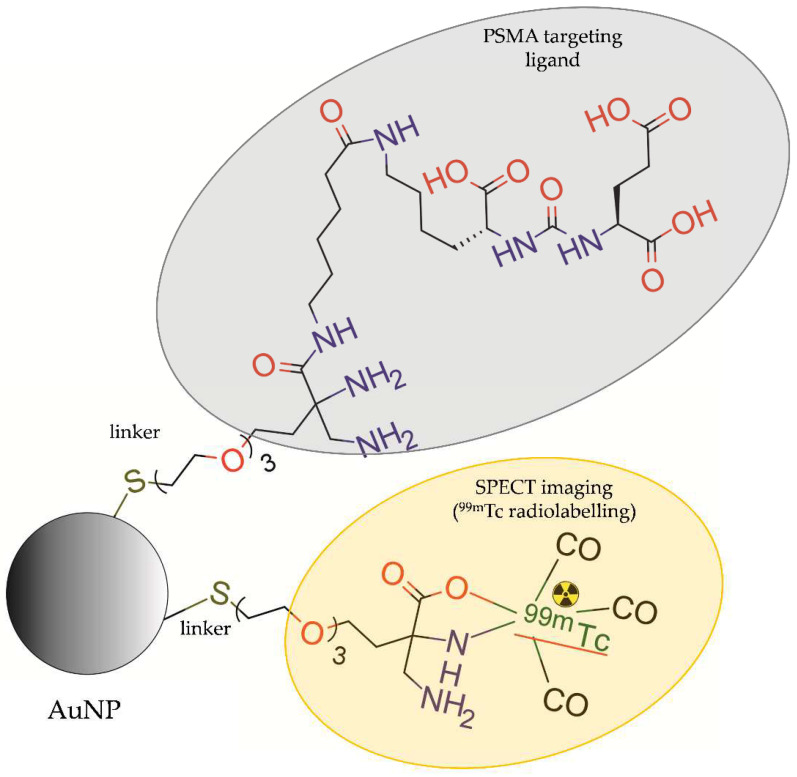
Structure of [^99m^Tc]DAP-PEG-AuNP-PEG-DAP-TF modified with coating ligands HS-PEG-DAP, functionalized with a small molecule urea-based PSMA-I inhibitor (TF), and radiolabelled with ^99m^Tc.

**Figure 13 molecules-28-04122-f013:**
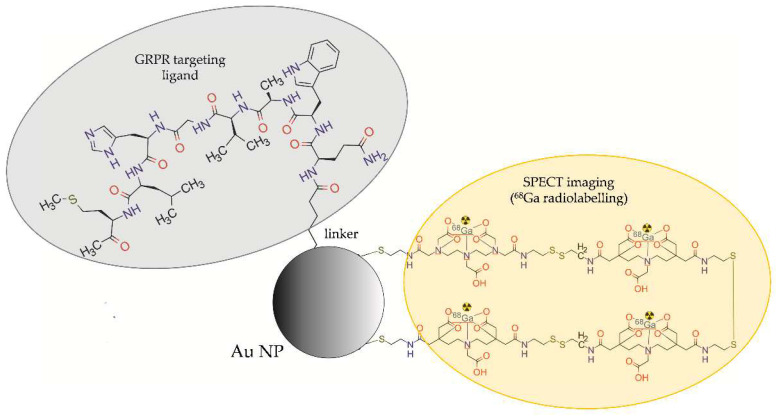
Structure of [^67^Ga]DTDTPA-AuNP-BBN radioconjugate consisting of AuNP modified with DTDTPA, functionalized with bombesin (BBN), and radiolabelled with ^67^Ga.

**Figure 14 molecules-28-04122-f014:**
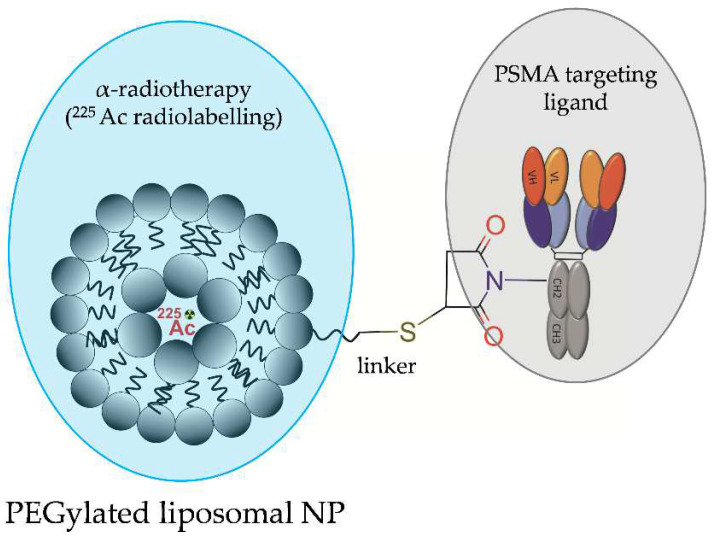
Structure of [^225^Ac]LNP-PEG-J591 radioconjugate consisting of PEG-ylated liposomal NP functionalized with anti-human PSMA J591 antibody and loaded with ^225^Ac.

**Figure 15 molecules-28-04122-f015:**
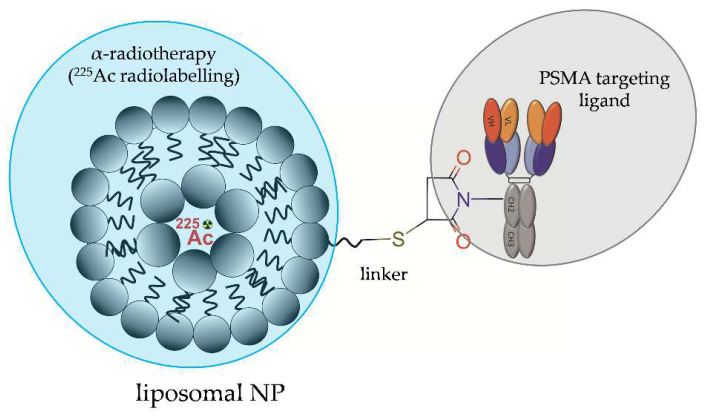
Structure of [^225^Ac]LNP-PEG-J591 radioconjugate consisting of liposome NP functionalized with anti-PSMA J591 antibody and loaded with ^225^Ac.

**Figure 16 molecules-28-04122-f016:**
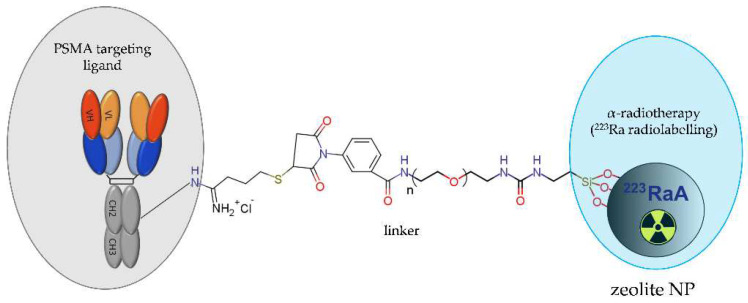
Structure of [^223^Ra]A-silane-PEG-D2B radioconjugate consisting of NaA zeolite NP loaded with ^223^Ra radionuclide, modified with silane-PEG molecules, and functionalized with the anti-PSMA D2B antibody.

**Figure 17 molecules-28-04122-f017:**
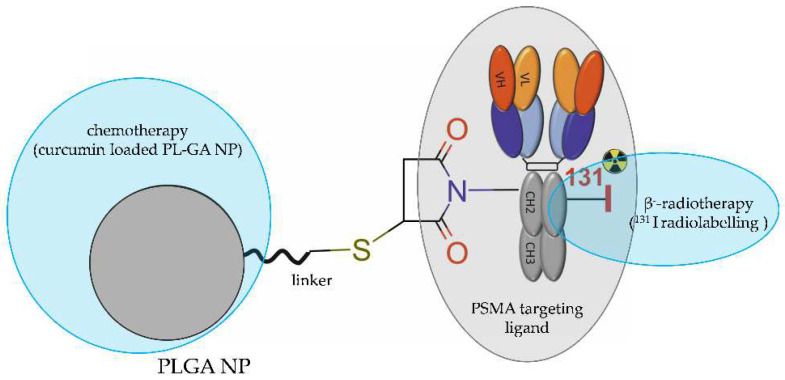
Structure of theranostic [^131^I]PSMA-PLGA-CUR radioconjugate consisting of PLGA NP loaded with curcumin, functionalized with the anti-PSMA J591 antibody, and radiolabelled with ^131^I.

**Figure 18 molecules-28-04122-f018:**
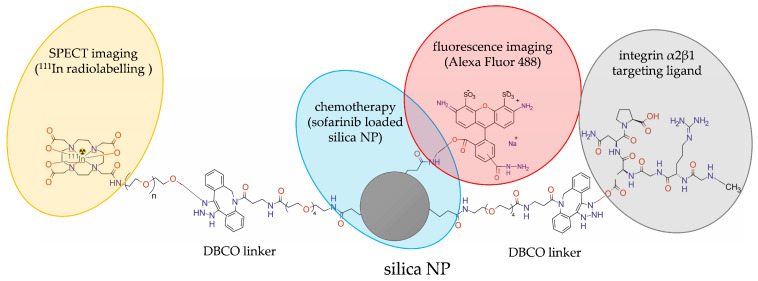
Structure of [^111^In]UnTHCPSi-Alexa488-DBCO-DOTA-iRGD-NP radioconjugate consisting of sofarinib loaded porous silica NP, modified with dibenzocyclooctyn (DBCO), functionalized with iRGD peptide, labelled with the fluorescent Alexa Fluor 488 dye, and radiolabelled with ^111^In.

**Figure 19 molecules-28-04122-f019:**
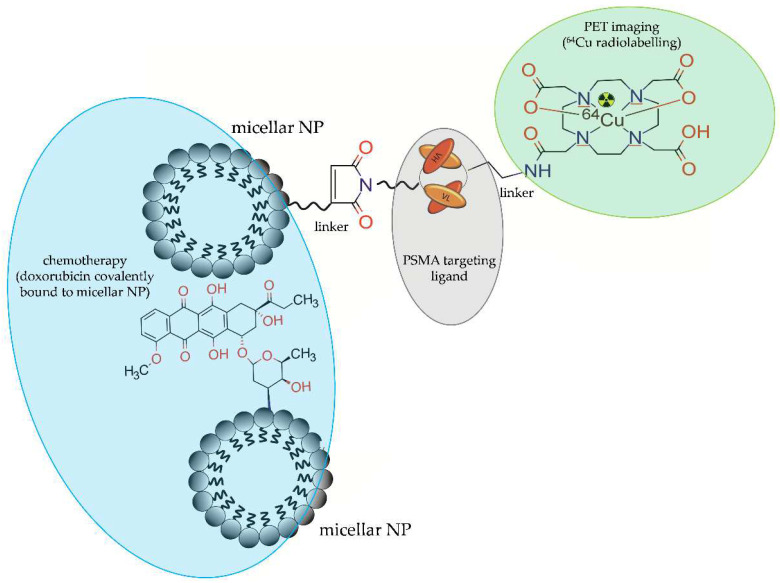
Mixture of doxorubicin-attached PEG micellar NP radiolabelled with ^64^Cu [^64^Cu]DOTA-Dox-PEG-LNP] and PEG micellar NP functionalized with anti-PSMA diabody (cys-DB) and radiolabelled with ^64^Cu [^64^Cu]DOTA-cys-DB-PEG-LNP.

**Figure 20 molecules-28-04122-f020:**
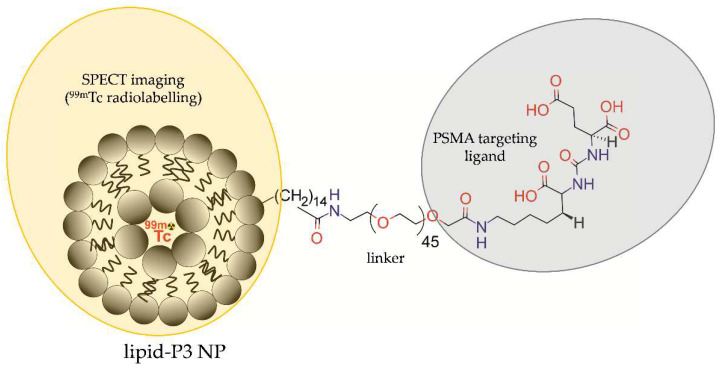
Structure of [^99m^Tc]P3-liposomes comprising liposomal NP modified by lipopolymer (P^3^) contained the small molecule Lys–urea–Glu-based PSMA inhibitor (PSMAL), PEG_2000_, and palmitate linker, and radiolabelled with ^99m^Tc.

**Figure 21 molecules-28-04122-f021:**
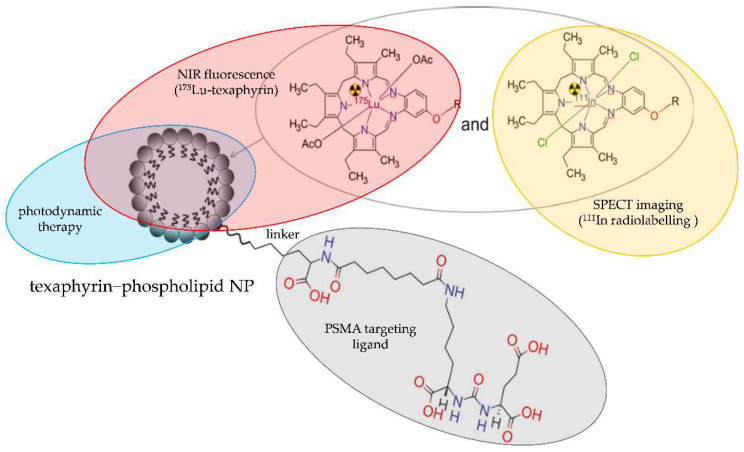
Structure of [^111^In/^175^Lu]-texaphyrin-PSMA radioconjugate comprising texaphyrin−phospholipid NP labelled with ^111^In and ^175^Lu and functionalized with urea-bearing PSMA-targeting YC-XII-35 moiety.

## Data Availability

The data presented in this study are available in Appendix A.

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
