# Peer review of "Nanoparticle-Based Radioconjugates for Targeted Imaging and Therapy of Prostate Cancer"

_molecules, 2023, doi:10.3390/molecules28104122_

Round 1
Reviewer 1 Report
In this review manuscript, the authors focus on presenting various nanoparticle based radioconjugates used for their theranostic applications in prostate cancer therapy and imaging. Since therapy and imaging are two important components in cancer therapy to treat and monitor the effect of treatment, this review presents various details on the application of radiolabelled nanoparticles in a systematic order. Specifically, the authors consolidate their review by providing selection criteria needed for the choice of nanoparticles, targeting ligands, radionuclides, and imaging methods, while addressing their clinical potential. Overall, the manuscript is written well and can be considered for publication after addressing the following major comments.
1. There are too many schematic figures (all 21 figures) showing the nanoparticles with various linkers and targeting ligands used in different studies. They are not very well balancing to support the focus of the review.
2. The authors should provide figures showing the nanoparticles-based multimodality imaging used in pre-clinical and clinical models of prostate cancers.
3. The review also superficially addresses the nanoconjugates used in different studies. The authors should discuss the advantages and disadvantages of the various systems reported previously.
4. Please provide a table showing various animal models used for theranostic imaging of prostate cancer.
5. Please also provide a table showing theranostic nanoparticles currently in clinical trials and approved for clinical use.
Reviewer 2 Report
Lankoff et al. summarized the nanoparticle-based radioconjugates for targeted imaging and therapy of prostate cancer in this review article. The article is well-structured and written. The article can be accepted for publication after minor corrections.
Comments:
1. Please include an outcome column in Table 1. This will immensely benefit the reader.
Reviewer 3 Report
Presented by Dr. Lankoff article is appealing to broad audience and provides significant insight into important application of radiolabeled nanoparticles for prostate cancer cure. Work is original and significant, all providen literature references are appropriate and up to date.
Additionally I would like to suggest add few paragraphs to conclusion, regarding the pipeline of drugs, passing last stage of clinical trials or registration and latest approved radiolabeled drugs for prostate cancer.
Minor issues:
Please provide the list of abbreviations
73 Provide additional links for 16,17
276 Provide additional references for 64Cu
287 Fig.1 Replace En with Eu
414 Please provide references
436 Please provide references
Reviewer 4 Report
In the paper Nanoparticle-based radioconjugates for targeted imaging and therapy of prostate cancer authors must see the following
1. Epidemalogy for the prostate cance should be given according to globocan.
2. Plant based nanoparticles or biogenically synthesized particle may also be given.
3. 1-2 lines for charectarization of nanoparticle should be given.
4. representative image or figure or generalised figure should be given.
5. Check for english grammar and spelling mistakes.
6. Check for references according to journal format.
Round 2
Reviewer 4 Report
authors have incorporated the comments